# The DREAM complex through its subunit Lin37 cooperates with Rb to initiate quiescence

Christina FS Mages[1], Axel Wintsche[1,2], Stephan H Bernhart[3,4], Gerd A Müller[1]*

[1]Molecular Oncology, Medical School, University of Leipzig, Leipzig, Germany; [2]Computational EvoDevo Group, Department of Computer Science and Interdisciplinary Center for Bioinformatics, University of Leipzig, Leipzig, Germany; [3]Transcriptome Bioinformatics Group, Department of Computer Science, University of Leipzig, Leipzig, Germany; [4]Interdisciplinary Center for Bioinformatics, University of Leipzig, Leipzig, Germany

**Abstract** The retinoblastoma Rb protein is an important factor controlling the cell cycle. Yet, mammalian cells carrying Rb deletions are still able to arrest under growth-limiting conditions. The Rb-related proteins p107 and p130, which are components of the DREAM complex, had been suggested to be responsible for a continued ability to arrest by inhibiting E2f activity and by recruiting chromatin-modifying enzymes. Here, we show that p130 and p107 are not sufficient for DREAM-dependent repression. We identify the MuvB protein Lin37 as an essential factor for DREAM function. Cells not expressing Lin37 proliferate normally, but DREAM completely loses its ability to repress genes in $G_0/G_1$ while all remaining subunits, including p130/p107, still bind to target gene promoters. Furthermore, cells lacking both Rb and Lin37 are incapable of exiting the cell cycle. Thus, Lin37 is an essential component of DREAM that cooperates with Rb to induce quiescence.

DOI: https://doi.org/10.7554/eLife.26876.001

*For correspondence: gerd.mueller@medizin.uni-leipzig.de

Competing interests: The authors declare that no competing interests exist.

## Introduction

The retinoblastoma protein Rb regulates cell cycle-dependent gene expression and is mutated in many cancer types (*Hanahan and Weinberg, 2000*). Rb and the related proteins p130 and p107 are members of the pocket protein family. All of them interact with E2f proteins to form complexes that repress expression of genes necessary for cell cycle progression in $G_0$ and early $G_1$ (*Classon and Dyson, 2001*; *Classon and Harlow, 2002*; *Cobrinik, 2005*; *Dick and Rubin, 2013*; *Burkhart and Sage, 2008*). Rb preferentially interacts with E2f1-3, and p130/p107 mainly bind to E2f4/5 (*Helin et al., 1992*; *Lees et al., 1993*; *Liban et al., 2017*). p107 or p130 are components of the DREAM complex (*Guiley et al., 2015*; *Litovchick et al., 2007*; *Pilkinton et al., 2007*; *Schmit et al., 2007*). While DREAM binds and represses cell cycle genes with maximal expression in S, $G_2$, and M phases through E2F and CHR promoter elements, E2f-Rb complexes cannot bind to CHR sites. However, they can interact with E2F binding sites of S phase genes (*Litovchick et al., 2007*; *Hurford et al., 1997*; *Müller et al., 2014*; *Müller et al., 2016*). Thus, Rb and p130/p107 have overlapping, but also clearly distinct functions in the repression of cell cycle genes.

Pocket protein triple knockout cells can proliferate, but are unable to exit the cell cycle under growth-limiting conditions (*Dannenberg et al., 2000*; *Sage et al., 2000*). In contrast, $Rb^{-/-}$ or $p130^{-/-}/p107^{-/-}$ cells largely maintain their potential to arrest in $G_0$ (*Hurford et al., 1997*; *Dannenberg et al., 2000*; *Sage et al., 2000*; *Herrera et al., 1996*). It was suggested that pocket proteins can substitute for each other in repressing E2f function and recruiting histone-modifying

enzymes to promoters of cell cycle genes. After it was discovered that p130 or p107 bind to cell cycle gene promoters as part of DREAM in $G_0/G_1$ (*Litovchick et al., 2007*; *Schmit et al., 2007*), it remained unclear whether MuvB components of DREAM contribute to the repressor function.

The MuvB core complex consists of Lin54, Lin52, Lin37, Lin9, and Rbbp4. The p130/p107-E2f4/5-Dp module is recruited to the MuvB core through a direct interaction of p130/p107 with Lin52 phosphorylated at Serine 28 (*Guiley et al., 2015*; *Litovchick et al., 2011*). Lin54 mediates binding of MuvB complexes to DNA through CHR promoter elements of $G_2/M$ cell cycle genes (*Marceau et al., 2016*; *Schmit et al., 2009*), and E2f4/5-Dp interact with E2F sites in promoters of $G_1/S$ genes. Because of its binding to E2F and CHR sites, DREAM is recruited to a broad set of cell cycle genes (*Litovchick et al., 2007*; *Müller et al., 2014*; *Müller et al., 2016*). Since Lin9 binds to several MuvB complex proteins (*Schmit et al., 2007*; *Wiseman et al., 2015*), it seems to be the central structural component of MuvB complexes. Rbbp4 can bind to histones and is involved in chromatin remodeling while being a component of other complexes like NuRD (*Tong et al., 1995*; *Zhang et al., 1998*), however, its exact function as part of MuvB complexes still has to be evaluated.

During progression through the cell cycle, p130/p107, E2f4/5, and Dp dissociate from MuvB. The MuvB core complex then interacts with B-myb and Foxm1 and switches its function from a transcriptional repressor to an activator (*Litovchick et al., 2007*; *Schmit et al., 2007*; *Sadasivam et al., 2012*). The B-myb-MuvB (MMB) complex forms in S phase, and is required for initial transcriptional activation and for recruiting Foxm1. Finally, the Foxm1-MuvB complex stimulates maximum expression of $G_2/M$ cell cycle genes (*Sadasivam et al., 2012*; *Chen et al., 2013*; *Down et al., 2012*). Mutation or reduced expression of Foxm1 or B-myb lead to decreased expression levels of $G_2/M$ genes followed by defects and cellular arrest during mitotis and cytokinesis (*Tarasov et al., 2008*; *Laoukili et al., 2005*; *Knight et al., 2009*). Similar observations were made for several MuvB proteins. Since they are components of the transcriptional repressor and activator complexes, depletion of Lin9, Lin52, or Lin54 leads to elevated cell cycle gene expression in $G_0/G_1$ (*Litovchick et al., 2007*), but also to reduced expression during S, $G_2$, and M followed by mitotic arrest (*Schmit et al., 2007*; *Knight et al., 2009*; *Boichuk et al., 2013*; *Kittler et al., 2007*; *Reichert et al., 2010*).

Lin37 is the only MuvB component without a defined role in transcription or generally in cell cycle regulation. Thus, we created and analyzed Lin37-deficient cells. We find that $Lin37^{-/-}$ cells proliferate normally. However, DREAM no longer acts as a transcriptional repressor in $G_0/G_1$. In contrast, activation of transcription by MMB and Foxm1-MuvB complexes still functions in Lin37-deficient cells. Interestingly, DREAM without Lin37 can still assemble and bind to promoters. Hence, recruitment of p130 or p107 pocket proteins is not sufficient for cell cycle gene repression. Furthermore, we show that $Lin37^{-/-}/Rb^{-/-}$ cells are essentially unable to exit the cell cycle. In conclusion, our data suggest that Lin37 as a part of DREAM cooperates with Rb in initiating quiescence.

## Results

### Lin37 is a nuclear protein that binds to MuvB complexes through two conserved domains

We aligned Lin37 protein sequences of several deuterostome species to identify evolutionary conserved regions that may exhibit specific functions. The sequence analysis revealed that Lin37 does not contain any conserved domains with known function. Lin37 is highly conserved in mammals, but poorly conserved in deuterostomes. However, three small conserved regions in the central part of the protein could be identified: a potential nuclear localization signal (NLS) and two adjacent conserved domains (CD1 and CD2) that are separated by a spacer of eight weakly conserved amino acids. To analyze the impact of the domains, we introduced point mutations changing the amino acid composition at two or three positions, respectively (*Figure 1A*, *Figure 1—figure supplement 1*). Given that Lin37 has been identified as a component of MuvB complexes in several species (*Sadasivam and DeCaprio, 2013*), we hypothesized that the interaction of Lin37 with the other MuvB core proteins may rely on the conserved domains CD1 or CD2. Indeed, mutation of CD1 or CD2 leads to reduced binding to Lin9, Lin54 and B-myb (*Figure 1B and C*). Binding was essentially lost when both domains were mutated simultaneously, providing evidence that both domains mediate binding of Lin37 to the MuvB core complex. GFP-tagged Lin37 was clearly localized to the nucleus, while mutation of the NLS resulted in a homogeneous protein distribution in the cell. In

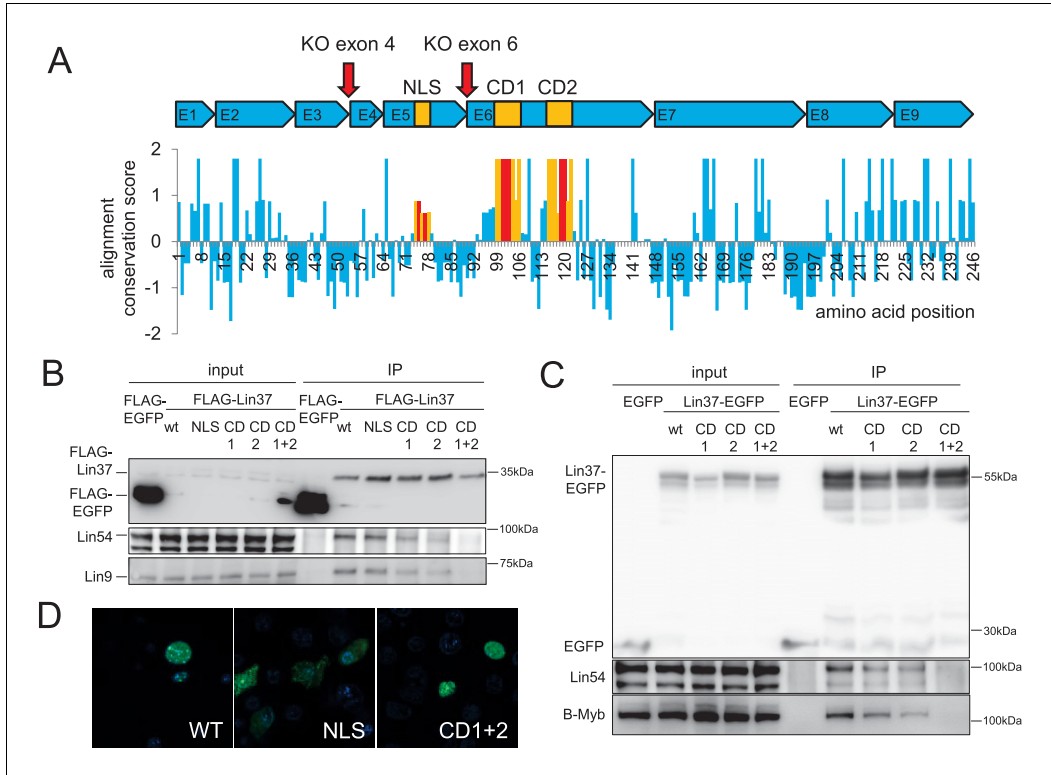

**Figure 1.** Conserved sequences in LIN37 are essential for nuclear localization and interaction with MuvB complex components. (**A**) Structure and evolutionary conservation of Lin37. Conserved domains are highlighted in yellow and introduced mutations in red (NLS: K75A/R77A, CD1: D101A/R102G/S103A, CD2: C119A/R120G). Red arrows mark target sites for CRISPR/Cas9 mutagenesis. Wild-type and mutant Lin37 fused with N-terminal FLAG tags or C-terminal EGFP tags were expressed in (**B**) NIH3T3 cells or (**C**) HCT116 cells (input, 10 µg whole cell extract) and precipitated with anti-Flag antibodies or anti-EGFP antibodies (IP). The MuvB complex components Lin9, Lin54 and B-Myb were tested for interaction with Lin37 wild-type (WT) and mutants (NLS, CD1, CD2, CD1 +2). (**D**) Fusions of Lin37 variants with EGFP were expressed in NIH3T3 cells and tested for their subcellular localization (green). Nuclei were stained with Hoechst33342 (blue).
DOI: https://doi.org/10.7554/eLife.26876.002
The following figure supplement is available for figure 1:

**Figure supplement 1.** Lin37 protein is weakly conserved in deuterostomes.
DOI: https://doi.org/10.7554/eLife.26876.003

contrast, mutation of CD1 and CD2 domains did not influence subcellular localization (*Figure 1D*). Thus, proper nuclear import of Lin37 depends on the NLS, but not on the interaction with MuvB.

## Knockout of Lin37 does not influence cell viability

To test whether Lin37 is essential for proliferation and cell survival similar to other MuvB components, we applied a CRISPR/Cas9 nickase approach to abrogate expression of the protein while minimizing off-target effects. Plasmids expressing sgRNAs targeting either the 5' ends of exon 4 or of exon 6 directly upstream of the MuvB interaction domain (*Figure 1A*) together with the Cas9 nickase were transiently transfected in NIH3T3 cells. Both targeting approaches led to ablation of Lin37 with a high frequency of 40–60% (*Figure 2—figure supplement 1*) showing that Lin37 is not essential for survival of these cells. To analyze the function of Lin37, we stably introduced episomal vectors for inducible expression of wild-type Lin37, the MuvB-binding mutant (CD1 +2) or luciferase into one of the exon 4 and exon 6 knockout cell lines that we had genotyped at the *Lin37* locus (*Figure 2—figure supplement 1*). Western Blot analysis revealed that Lin37 can be re-expressed after induction with doxycycline. Given that wild-type Lin37 was already detected in un-induced cells (*Figure 2A*), we refrained from comparing un-induced with induced cells. Alternatively, we analyzed differences

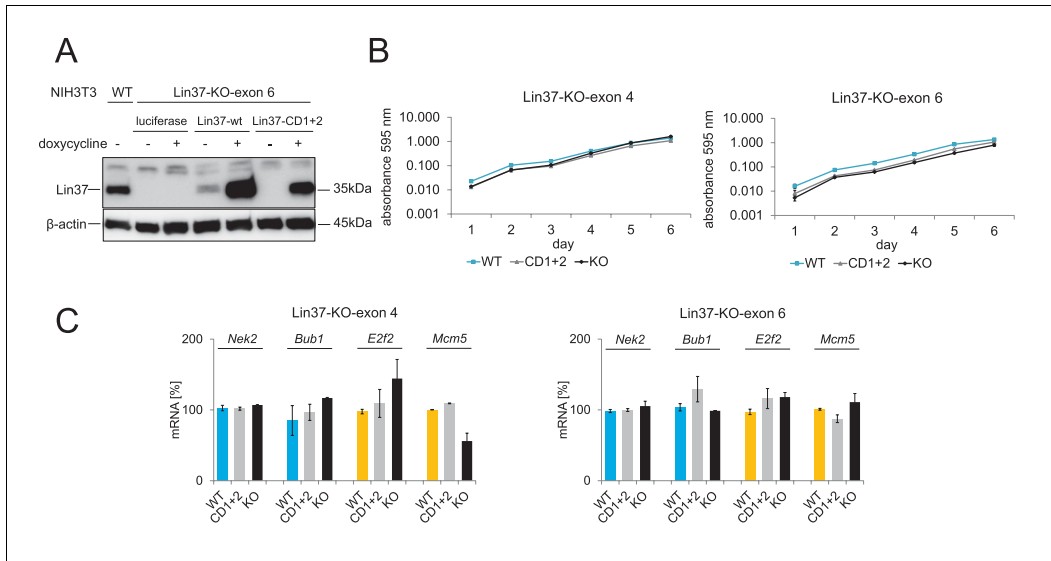

**Figure 2.** Cell viability and mRNA expression of early and late cell cycle genes in proliferating cells do not depend on Lin37. Episomal constructs for inducible expression of Lin37 (WT), the MuvB-binding mutant (CD1 +2) and luciferase (KO) were stably introduced in Lin37 knockout clones mutated in exon 4 or exon 6. (**A**) Lin37-KO-exon 6 cells were tested for doxycycline-induced expression by Western blot. (**B**) The viability of two independent Lin37 knockout cell lines, generated by mutation of either exon 4 or exon 6, was tested by MTT assays for 6 days after stable transfection with episomal rescue constructs. Mean values ± SD of three biological replicates are shown. (**C**) Expression of $G_2/M$ (blue) and $G_1/S$ (yellow) cell cycle genes in proliferating cells was analyzed by qPCR. Mean values ± SD of one representative experiment for all cell lines with two technical replicates are given. All data points for each gene were normalized to the mRNA expression of the first replicate in wild-type cells which was set to 100%.

DOI: https://doi.org/10.7554/eLife.26876.004
The following figure supplement is available for figure 2:

**Figure supplement 1.** Lin37 knockout clones can be established by CRISPR/Cas9 nickase with a high efficiency.
DOI: https://doi.org/10.7554/eLife.26876.005

between cell lines induced for the expression of wild-type Lin37, the MuvB binding deficient mutant (CD1+2), or a luciferase negative control (KO).

To test the influence of Lin37 on the viability of the cell lines, we performed MTT assays. The results indicate that viability of the cell lines does not depend on the expression of wild-type Lin37 (*Figure 2B*). Next, we tested whether expression of early and late cell cycle genes is influenced by the Lin37 status of proliferating cells. Expression of *E2f2* and *Mcm5*, which display peak expression during S phase, and *Bub1* and *Nek2*, which are mainly expressed during $G_2/M$, were essentially unaffected by Lin37 (*Figure 2C*). Thus, the data suggest that loss of Lin37 does not interfere with the proliferation rate or gene activation by B-myb-MuvB/Foxm1-MuvB complexes.

## Lin37 knockout leads to transcriptional activation of cell cycle genes in $G_0/G_1$

Next, we tested the hypothesis that Lin37 knockout only affects DREAM repressor and not MuvB-based activator function. We synchronized several independent Lin37 wild-type (n = 5) and knockout (n = 6) clones by serum starvation and re-stimulated the cells for 9 hr, 18 hr, and 27 hr to obtain cell populations enriched for $G_0$, $G_1$, S, and $G_2/M$ phases. Lin37 knockout cell lines maintain the potential to arrest in $G_0$, although a minor increase in the fraction of S and $G_2/M$ phase cells was observed. Furthermore, the number of cells in S and $G_2/M$ phases was slightly elevated at the 18 hr time point (*Figure 3A*). mRNA expression analysis of early and late cell cycle genes revealed a significant de-repression in $G_0$ and $G_1$. In S and $G_2/M$ phases, expression of both gene groups was still moderately elevated when Lin37 was absent or mutated (*Figure 3B*). These results show that Lin37 is required

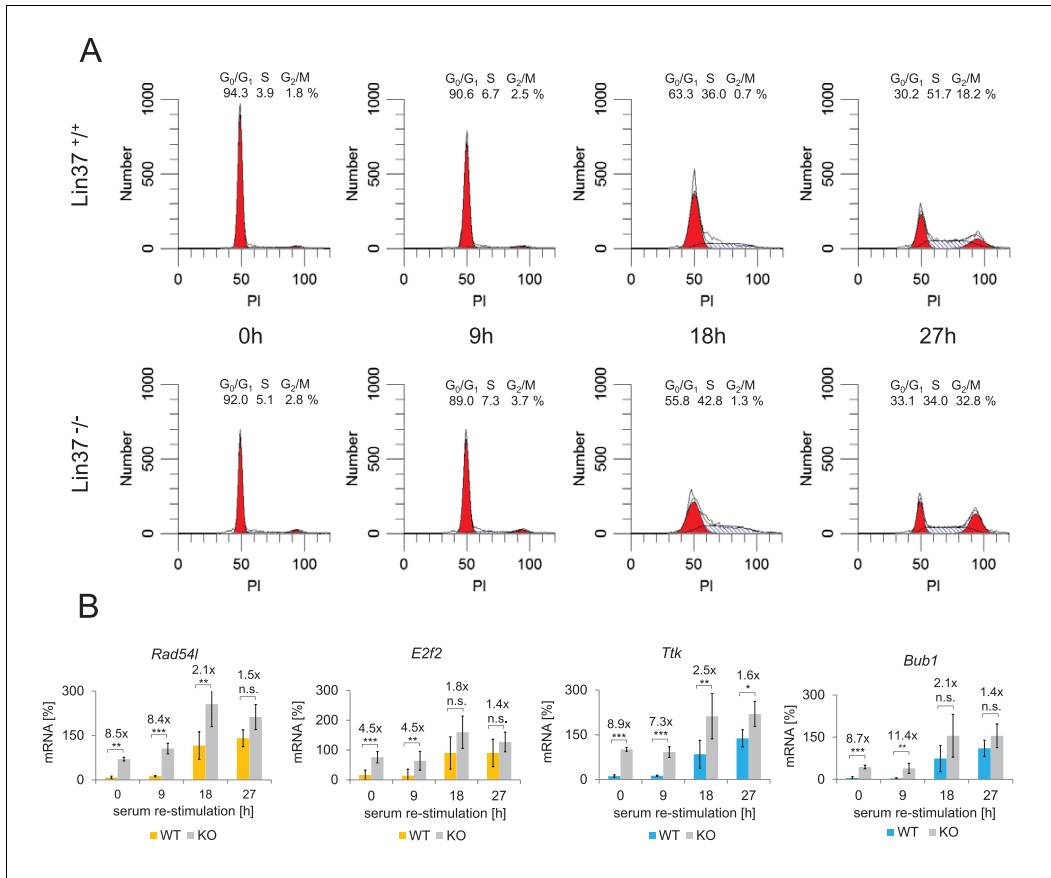

**Figure 3.** Knockout of Lin37 results in a de-repression of cell cycle genes in $G_0/G_1$. The NIH3T3 parental cell line as well as independent wild-type (WT, n = 4) and Lin37 knockout (KO, n = 6) single cell clones were arrested in $G_0$ by serum deprivation and re-stimulated for 9, 18, and 27 hr. (**A**) Cell cycle distribution was measured by PI staining and flow cytometry. One representative example of wild-type ($Lin37^{+/+}$) and knockout ($Lin37^{-/-}$) cells is shown. (**B**) mRNA of $G_1/S$ (yellow) and $G_2/M$ (blue) cell cycle genes was measured by qPCR in all cell lines with two technical replicates each and normalized to U6 expression. All data points for each gene were further normalized to the maximum mRNA expression of the parental cell line which was set to 100%. Mean values ± SD are given, and significances were calculated by the Students T-Test (*p≤0.05, **p≤0.01, ***p≤0.001).
DOI: https://doi.org/10.7554/eLife.26876.006

for transcriptional repression by DREAM but not for activation by B-myb-MuvB and Foxm1-MuvB, which is intriguing because Lin37 is a component of both the repressive and activating complexes.

## Ablation of Lin37 leads to a complete loss of DREAM function

Having established that knockout of Lin37 results in an up-regulation of cell cycle gene mRNA expression in serum-starved cells, we next asked whether DREAM completely loses its function as a transcriptional repressor without Lin37 or whether DREAM can maintain partial activity. To this end, we performed luciferase reporter assays with promoters of known DREAM target genes. In serum-starved NIH3T3 wild-type cells, activity of DREAM target promoters was up-regulated when the E2F binding sites (*Orc1*, *Cdc45*) or the CHRs (*Ttk*, *Ccnb2*) were mutated (*Figure 4*). Because DREAM cannot bind and repress these promoters anymore (*Müller et al., 2016*), the increase in activity reflects complete loss of DREAM function at each specific promoter. Notably, mutating the CHR in the *Ttk* promoter led to an increase in activity of merely 2-fold in knockout cells compared to 136-fold in Lin37 wild-type cells. Furthermore, the *Ccnb2* CHR mutant promoter displayed even slightly less activity than the wild-type promoter in $Lin37^{-/-}$ cells. Activities of *Cdc45* wild-type and E2F-site mutant promoters were not significantly different in Lin37 knockout cells, while mutating the E2F binding site in the *Orc1* promoter still increased activity 2.7-fold in the absence of Lin37.

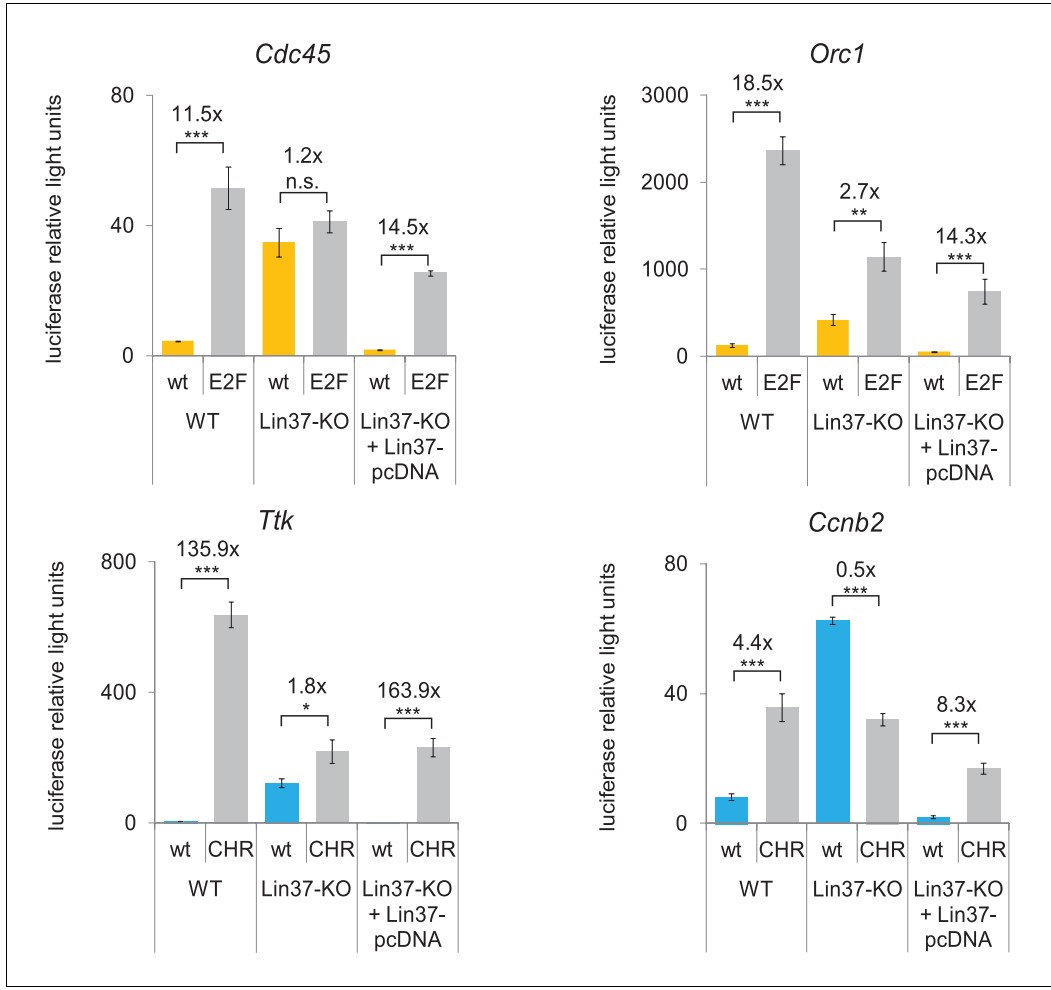

**Figure 4.** Knockout of Lin37 leads to a de-repression of cell cycle gene promoters in quiescence. Wild-type (WT) or Lin37 knockout cells (Lin37-KO; clone 632–2) were transfected with luciferase reporter constructs of $G_1/S$ (yellow) and $G_2/M$ (blue) cell cycle gene promoters. Wild-type (wt) and DREAM-binding site deficient (E2F, CHR) promoters were tested for their activity. Lin37 knockout was rescued by re-expression of Lin37 (Lin37-pcDNA). Cells were arrested by serum deprivation. Promoter activities were normalized to Renilla luciferase activity. Mean values ± SD of three biological replicates are given, and significances were calculated by the Students T-Test ($^*p \leq 0.05$, $^{**}p \leq 0.01$, $^{***}p \leq 0.001$).
DOI: https://doi.org/10.7554/eLife.26876.007

Importantly, repression of all wild-type promoters could completely be restored when a Lin37 expression plasmid was co-transfected (*Figure 4*). As mutation of CHR sites, which exclusively bind DREAM but not E2F/Rb in repressed promoters, did not significantly increase gene expression in Lin37 knockout cells, we conclude that activity of DREAM is completely lost in these cells. We propose that the small but significant increase of *Orc1* activity upon E2F site mutation relates to E2f/Rb repressor complexes that bind to E2F elements and are still active in Lin37 knockout cells. Together, these results strongly suggest that DREAM is essential for repression of cell cycle genes through CHR and E2F sites in quiescence and that loss of DREAM activity is not compensated by other repressor complexes.

## Expression of a Lin37 MuvB-binding mutant cannot rescue the knockout phenotype

To test whether binding of Lin37 to DREAM is essential for its function in cell cycle gene regulation, we performed time-course experiments with serum-starved and re-stimulated knockout cells that

were stably transfected with episomes expressing Lin37 (WT), the MuvB binding-deficient mutant (CD1+2) or luciferase (KO) (*Figure 5A*, *Figure 5—source data 1*). mRNA levels of $G_2/M$ (*Bub1*, *Nek2*) and $G_1/S$ (*E2f2*, *Mcm5*) cell cycle genes were clearly increased during $G_0$ and early $G_1$ in cell lines expressing luciferase and the Lin37 mutant compared to cells expressing wild-type Lin37. In

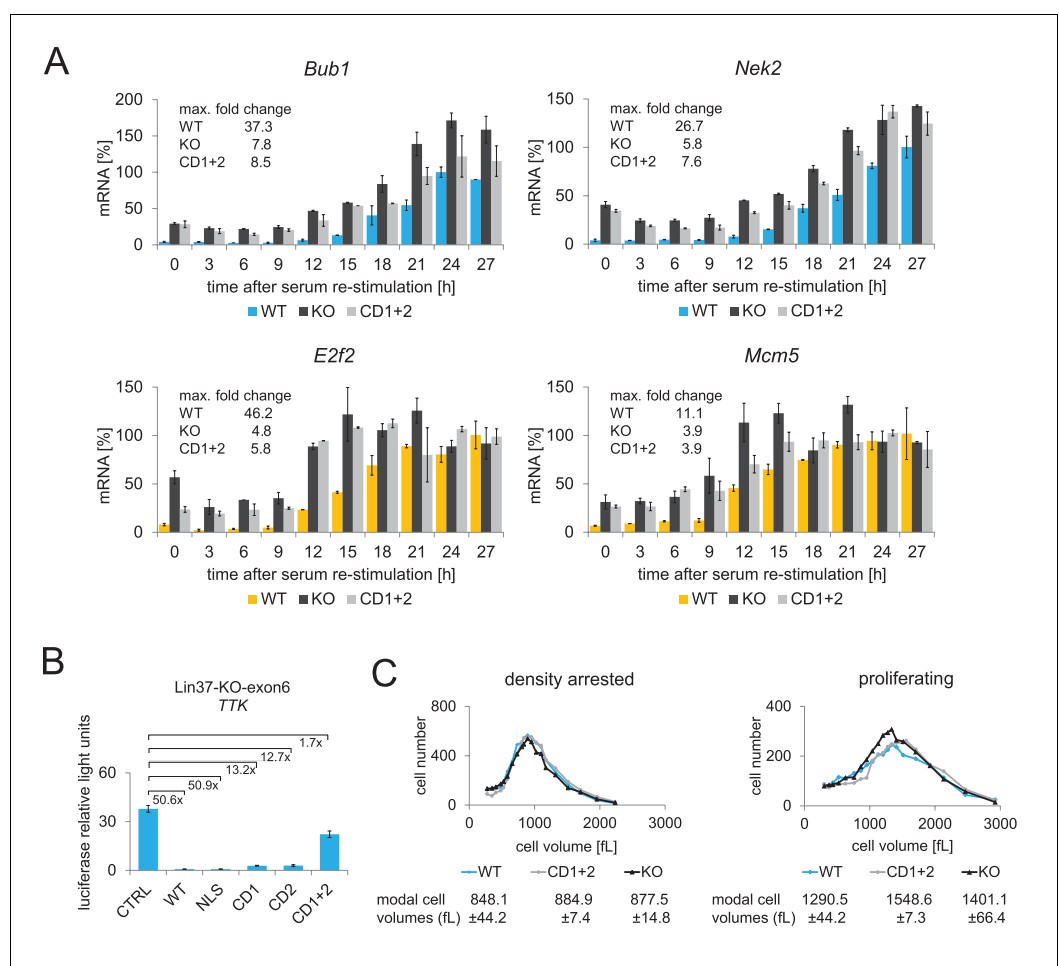

**Figure 5.** Expression of wild-type Lin37, but not of a MuvB-binding deficient mutant, rescues the knockout phenotype. (**A**) Lin37 knockout cells stably transfected with episomal vectors expressing wild-type Lin37 (WT), a MuvB-binding deficient mutant (CD1+2) or luciferase (KO) were arrested by serum starvation in $G_0$ and stimulated to re-enter the cell cycle. mRNA expression of $G_2/M$ (blue) and $G_1/S$ (yellow) cell cycle genes was measured by qPCR at given time points after serum re-stimulation. All data points for each gene were normalized to the maximum mRNA expression of the Lin37 wild-type rescue cell line which was set to 100%. Mean values ± SD of one representative experiment with two technical replicates for each time point are shown. (**B**) Activity of the *Ttk* $G_2/M$ cell cycle gene promoter in serum-starved Lin37 knockout cells (clone 632–2) was determined by luciferase reporter assays while expressing wild-type Lin37 (WT), a mutant with non-functional nuclear localization signal (NLS), or mutants with altered MuvB binding domains (CD1, CD2, CD1+2). Mean values ± SD of three biological replicates are given and compared with *Ttk* promoter activity in Lin37 deficient cells (CTRL). (**C**) The volumes of density-arrested and proliferating Lin37-deficient cells (clone 632–2) expressing wild-type Lin37 (WT), a MuvB-binding deficient mutant (CD1+2) or luciferase (KO) was determined by Coulter counter analyses. Approximately 30,000 cells were measured in two biological replicates with three technical replicates each. One representative experiment is shown and the modal cell volumes ± SD are given.

DOI: https://doi.org/10.7554/eLife.26876.008

The following source data is available for figure 5:

**Source data 1.** Cell cycle analysis of synchronized *Lin37⁻/⁻* NIH3T3 expressing wild-type Lin37 (WT), a non-MuvB-binding Lin37 mutant (CD1+2) or luciferase (KO).

DOI: https://doi.org/10.7554/eLife.26876.009

contrast, this increase in mRNA expression was not or only to a minor extent detected in late cell cycle phases, leading to a reduced maximum fold-change of all genes measured (*Figure 5A*). This phenotype is consistent with the data shown for several single cell knockout clones (*Figure 3*). Thus, downregulation of mRNA expression in $G_0$ and early $G_1$ can be rescued by re-expression of wild-type Lin37 but not by MuvB binding-deficient mutant Lin37.

The cell cycle distributions of all samples were analyzed by PI-DNA staining and flow cytometry. Cells expressing no or mutated Lin37 showed a minor increase of cells in S and $G_2$/M phases after the serum deprivation period and entered S and $G_2$/M phases earlier than cells expressing wild-type Lin37 (*Figure 5—source data 1*). Additionally, we performed luciferase reporter assays measuring activity of the *Ttk* wild-type promoter in serum-starved Lin37 knockout cells when Lin37 wild-type protein or several mutants were expressed. Lin37 lacking a functional NLS retained its ability to fully restore DREAM function (*Figure 5B*). CD1 and CD2 mutants could rescue DREAM function, but less efficiently than wild-type Lin37, which is consistent with their reduced affinity for theMuvB complex (*Figure 1B and C*). In contrast, the CD1+2 double mutant does not interact at all with the MuvB core and was unable to restore repression of the *Ttk* promoter (*Figure 5B*).

Taken together, our results provide evidence that binding of Lin37 to DREAM is essential for repression of cell cycle genes.

It has been shown that over-expression of cyclin D1 leads to a reduced volume of proliferating and contact inhibited NIH3T3 cells (*Quelle et al., 1993*). As the activity of cyclin D-CDK complexes results in an increased phosphorylation of pocket proteins and a reduced activity of the DREAM repressor complex, we chose to test whether Lin37-deficient cells also exhibit this 'wee' phenotype. However, we found that Lin37 knockouts and cells expressing the MuvB-binding deficient mutant appear to be slightly larger than cells expressing wild-type Lin37 (*Figure 5C*), but these changes in the cell volume were not significant. Thus, loss of Lin37 does not reduce the cell volume as over-expression of cyclin D1 does.

## Transcriptome-wide identification of Lin37-regulated genes

To identify genes de-regulated after Lin37 knockout on a genome-wide level, we isolated RNA from serum-starved *Lin37*$^{-/-}$ cells either transfected with episomal vectors expressing luciferase (KO) or wild-type Lin37 (WT). One technical and two biological replicates were performed, which led to the identification of 387 significantly up- and 76 significantly down-regulated genes (*Supplementary file 2*). Gene Ontology (GO) analysis showed a highly significant enrichment of cell cycle-related processes for the up-regulated genes (*Figure 6A*, *Supplementary file 2*), including processes connected to S phase (e.g. DNA replication, DNA metabolic process) and to mitosis (e.g. chromosome segregation, mitotic nuclear division). In contrast, GO analysis yielded sets of developmental and differentiation processes for genes repressed after Lin37 knockout, but with much lower significances (*Supplementary file 2*). We then mapped the up-regulated genes identified in mouse cells to their human orthologues to compare them with datasets of DREAM targets and cell cycle genes (*Müller et al., 2014*). 77.5% of the top 40 de-repressed protein-coding genes (*Figure 6B*) have been identified as DREAM targets by ChIP-chip (*Litovchick et al., 2007*). 65% of the top hits are cell cycle genes with an expression maximum in $G_2$ and mitosis and contain evolutionary conserved CHR promoter elements close to their transcription start sites. Additionally, expression of more than one third of cell cycle genes de-repressed in *Lin37*$^{-/-}$ cells displayed peak expression in S phase (*Figure 6C*) and can bind DREAM though E2F sites (*Supplementary file 2*). Thus, both $G_2$/M and $G_1$/S cell cycle genes were affected by loss of Lin37. Interestingly, expression of four of the MuvB complex components – Foxm1, B-myb (Mybl2), Lin9, and Lin54 – were also up-regulated in Lin37 knockout cells (*Figure 6B*). Furthermore, analysis of all Lin37-regulated genes revealed that 157 of their human orthologues are bound by DREAM and differentially expressed in the cell cycle, while 42 and 43 are identified exclusively as DREAM targets or cell cycle genes, respectively. In contrast, several of the Lin37–regulated genes were not experimentally validated DREAM targets or cell cycle genes (*Figure 6D*). However, 60 of the 115 Lin37 targets without experimentally validated cell cycle expression have been predicted to be cell cycle genes (www.targetgenereg.org) (*Fischer et al., 2016*), suggesting that the actual overlap between Lin37 targets and cell cycle genes may be larger. In contrast to the genes up-regulated in *Lin37*$^{-/-}$ cells, none of the down-regulated genes bind DREAM and only four were identified as cell cycle genes (*Supplementary file 2*). Together, the

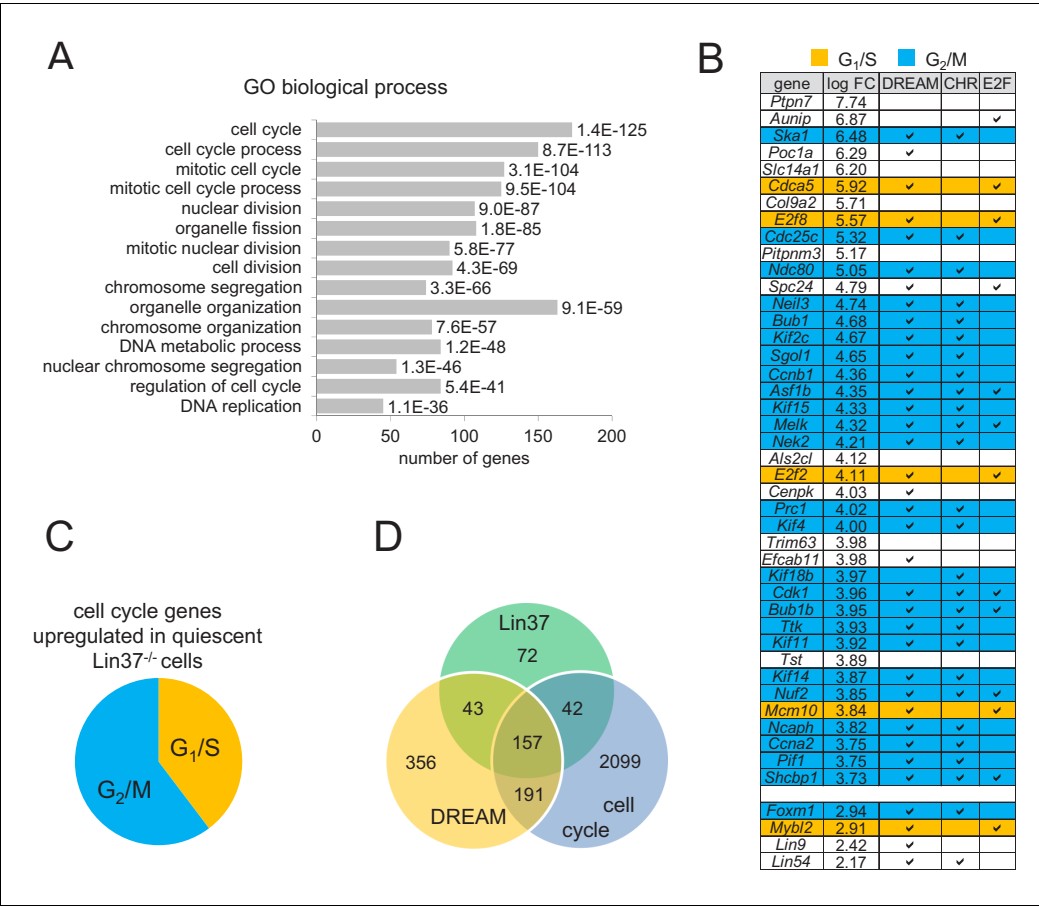

**Figure 6.** Expression of early and late cell cycle genes is up-regulated in quiescent *Lin37*$^{-/-}$ cells on a global level. (A) Genes up-regulated in quiescent *Lin37*$^{-/-}$ cells were analyzed for enrichment of Gene Ontology (GO) terms. The top fifteen overrepresented GO terms and the number of genes are given according to the p-value. For a complete list see *Supplementary file 2*. (B) Top 40 genes up-regulated (log fold change) in quiescent *Lin37*$^{-/-}$ cells and de-repressed genes coding for MuvB complex components are shown. Blue: Maximum expression in G$_2$/M. Yellow: Maximum expression in G$_1$/S. White: No experimentally validated differential cell cycle expression. A '✓' marks genes bound by DREAM components as detected by ChIP-on-chip analysis (*Litovchick et al., 2007*) and marks the presence of evolutionary conserved CHR or E2F binding sites close to the transcription start sites (*Müller et al., 2014*). For a complete list of all Lin37-regulated genes see *Supplementary file 2*. (C) Distribution of early and late cell cycle genes in all cell cycle genes regulated by Lin37. (D) Correlation of genes identified as regulated by Lin37, bound by DREAM and being differentially expressed during the cell cycle (*Müller et al., 2014*).

DOI: https://doi.org/10.7554/eLife.26876.010

transcriptome-wide analysis reveals that DREAM targets and cell cycle genes are generally up-regulated when Lin37 is not expressed.

## DREAM components assemble in the absence of Lin37

In order to test whether or not DREAM can form without Lin37, we performed DNA pulldown assays with a probe of the *Ccnb2* promoter containing a CDE/CHR element necessary for recruiting all DREAM components (*Müller et al., 2012*). DREAM-specific proteins were purified from both wild-type and from Lin37 knockout nuclear extracts (*Figure 7A*). These findings show that DREAM can assemble at CHR sites of DNA probes *in vitro* even in the absence of Lin37. To test for DREAM binding to cell cycle gene promoters *in vivo*, we performed chromatin immunoprecipitations (ChIP) with density-arrested wild-type and Lin37-deficient cells followed by qPCR. Binding signals of Lin37 at G$_1$/S and G$_2$/M cell cycle gene promoters were dramatically decreased in *Lin37*$^{-/-}$ cells, and binding

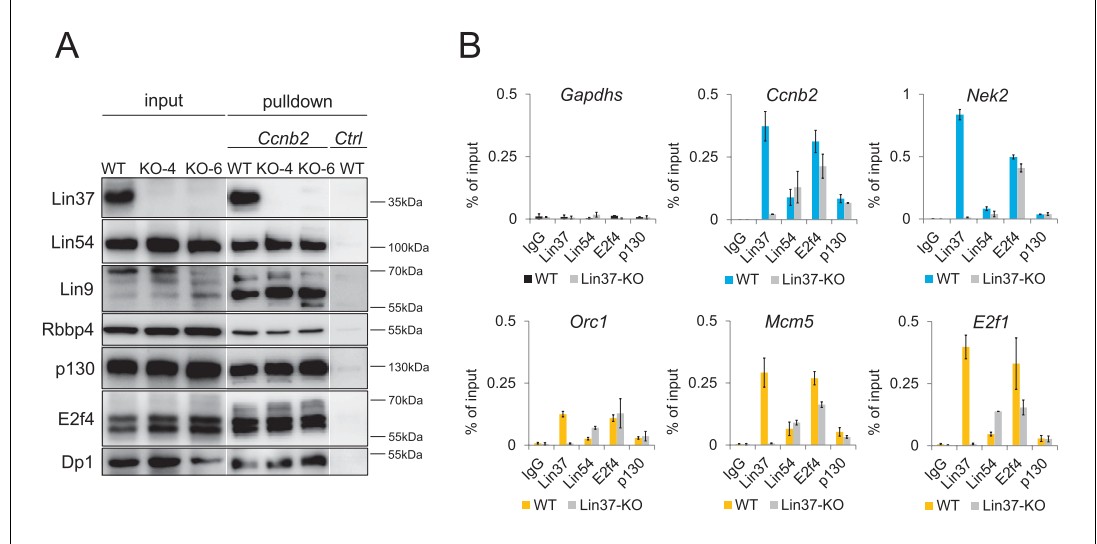

**Figure 7.** Lin37 is not essential for DREAM assembly at promoters. (A) DREAM components were purified from nuclear extracts of two independent density-arrested Lin37 knockout cell lines created by targeting exon 4 (KO-4) or exon 6 (KO-6) and NIH3T3 cells expressing Lin37 (WT). Purification was performed with a fragment of the mouse *cyclin B2* promoter (*Ccnb2*) containing a CDE/CHR tandem element or a fragment of the *Gapdhs* promoter without CHR or E2F sites (Ctrl) to determine background binding. Protein binding was analyzed by Western blotting. All samples detected with one specific antibody were run on the same gel. (B) *In vivo* binding of DREAM to $G_2/M$ (blue) and $G_1/S$ (yellow) cell cycle gene promoters in the Lin37$^{+/+}$ NIH3T3 parental cell line (WT) and a Lin37$^{-/-}$ cell line (Lin37-KO; clone 632-2) was analyzed by chromatin immunoprecipitation followed by semi-quantitative PCR (ChIP-qPCR). Enrichment of DREAM components normalized to input DNA is given. Non-targeting IgG and analysis of the non-DREAM binding *Gapdhs* promoter served as negative controls. Mean values ± SD of one representative experiment with three technical replicates are shown.

DOI: https://doi.org/10.7554/eLife.26876.011

of the DREAM components Lin54, E2f4, and p130 was only slightly or not altered (*Figure 7B*). The results obtained with both experimental approaches show that DREAM components assemble, but form a non-functional complex when Lin37 is absent.

## Loss of Lin37 leads to de-repression of cell cycle genes in quiescent C2C12 myoblast cells

To analyze whether the Lin37 knockout phenotypes described in NIH3T3 fibroblasts can be reproduced in a different cellular system, we created Lin37-deficient C2C12 myoblast cell lines with the same CRISPR/Cas9-nickase constructs used for generating NIH3T3 knockouts. Five Lin37-deficient single cell clones were selected for further analysis and compared with the C2C12 parental cell line and 4 clones that were treated with CRISPR/Cas9-nickase constructs but remained wild-type for Lin37 (*Figure 8A*). Flow cytometry revealed that the percentages of cells in $G_1$, S, and $G_2/M$ phases of the cell cycle did not significantly change either in proliferating or in serum-starved cells (*Figure 8B*). Thus, Lin37-deficient C2C12 cells proliferate normally and can arrest under growth limiting conditions. However, as observed in fibroblasts, knockout of Lin37 lead to a loss of DREAM function in quiescent C2C12 cells. While the expression of early and late cell cycle genes remained basically unchanged between proliferating wild-type and knockout cells, expression was highly increased in serum-starved knockout cells (*Figure 8C*). Next, we wanted to examine whether de-repression of cell cycle genes during quiescence can be rescued by re-expression of Lin37. To this end, we analyzed the activities of wild-type or DREAM binding site-deficient promoter reporter constructs of *Orc1* and *Ttk* in serum-starved C2C12 cells with or without transient expression of Lin37 (*Figure 8D*). As expected, activities of the *Orc1* and *Ttk* promoters increased in the C2C12 parental cells when the E2F or CHR sites were mutated. In three Lin37-deficient C2C12 cell lines, the differences in the activities of wild-type and mutant promoters were decreased. As observed in NIH3T3

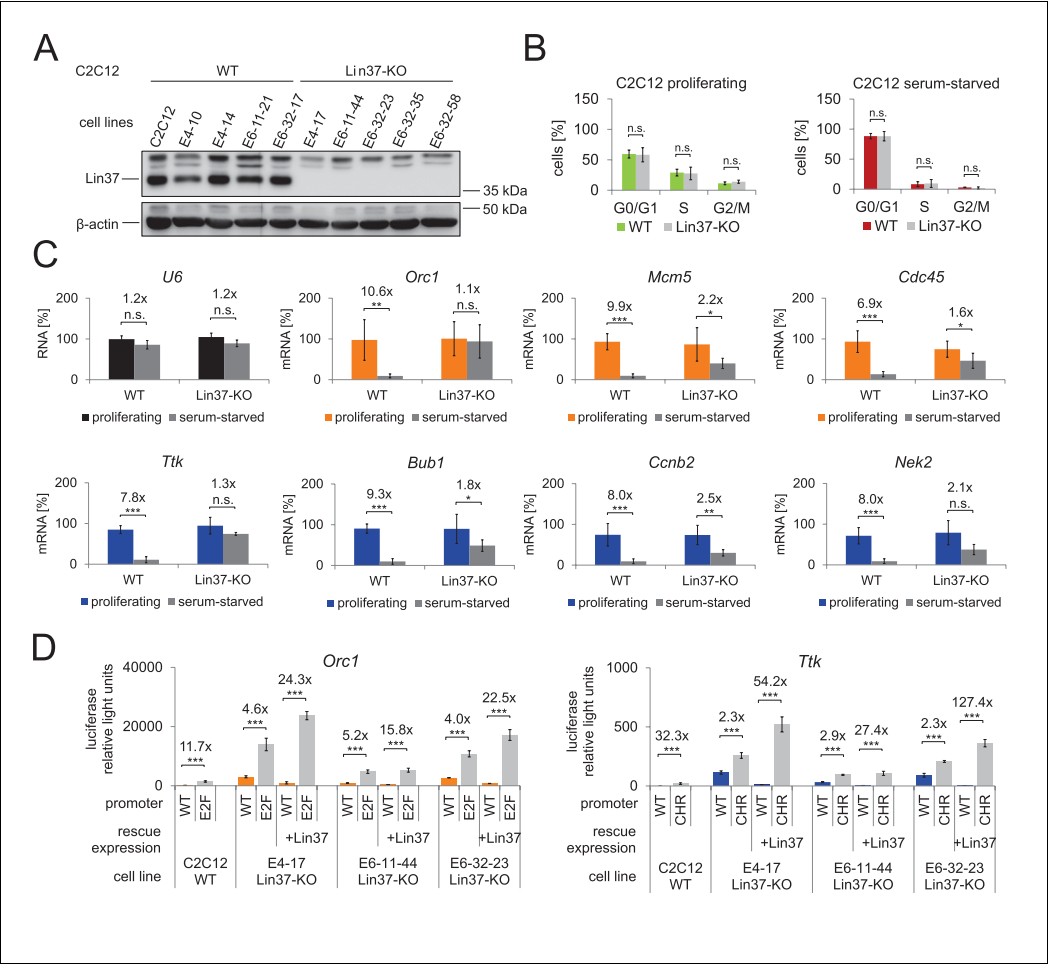

**Figure 8.** Knockout of Lin37 in C2C12 myoblast cells de-represses cell cycle genes in quiescence. (**A**) Western blot analysis of Lin37 wild-type C2C12 parental cells and single cell clones treated with plasmids expressing Cas9 nickase and sgRNAs targeting the 5' end of Lin37 exon 4 or 6. Cell extracts were tested for expression of Lin37 and β-actin. The five wild-type (WT) and Lin37 knockout (Lin37-KO) lines were utilized for further phenotypic analysis. (**B**) Lin37 wild-type (WT, n = 5) and Lin37 knockout (Lin37-KO, n = 5) lines were arrested in $G_0$ by serum deprivation. Cell cycle distribution of proliferating and quiescent cells was measured by PI staining and flow cytometry. (**C**) mRNA expression of $G_1$/S (orange) and $G_2$/M (blue) cell cycle genes was measured by qPCR in all cell lines with two technical replicates each and normalized to U6 expression. All data points for each gene were normalized to the mRNA expression of the proliferating parental cell line which was set to 100%. (**D**) Serum-starved wild-type (WT) or Lin37 knockout cell lines (Lin37-KO) were transfected with luciferase reporter constructs of the *Orc1* (G1/S) and *Ttk* (G2/M) cell cycle gene promoters. Wild-type (WT) and DREAM-binding site-deficient (E2F, CHR) promoters were tested for their activity. Lin37 knockout was rescued by re-expression of Lin37 (+Lin37). Promoter activities were normalized to Renilla luciferase activity. For all experiments shown in (**B**), (**C**), and (**D**), mean values ± SD are given, and significances were calculated by the Students T-Test (*p≤0.05, **p≤0.01, ***p≤0.001).

DOI: https://doi.org/10.7554/eLife.26876.012

cells, these effects were more pronounced when the *Ttk* promoter was analyzed, given that E2f/Rb complexes can bind to the E2F site of *Orc1* and partially repress the promoter in Lin37 knockout cells. However, re-expression of Lin37 completely rescued DREAM function in all cell clones.

Taken together, the data from C2C12 myoblast cells reflect the results obtained with NIH3T3 cells, which suggests that the Lin37 knockout phenotype is a general phenomenon.

## Combined knockout of Lin37 and Rb deregulates cell cycle gene expression and leads to a loss of cell cycle control

It has been shown that triple knockout of all pocket proteins leads to an inability of cells to enter quiescence in response to serum deprivation and contact inhibition (*Dannenberg et al., 2000*; *Sage et al., 2000*; *Classon et al., 2000*). As p130/p107 function is mostly achieved through DREAM (*Guiley et al., 2015*; *Litovchick et al., 2007*; *Pilkinton et al., 2007*; *Schmit et al., 2007*; *Forristal et al., 2014*), we hypothesized that this phenotype may also be observed when both Lin37 and Rb are inactive. To test this hypothesis, we created Rb single and Lin37/Rb double knockout NIH3T3 cells by targeting either exon 1 or exon 13 of the *Rb* gene in Lin37 positive or negative cells. $Rb^{-/-}$ and $Lin37^{-/-}/Rb^{-/-}$ cell clones could be isolated with both approaches (*Figure 9A*). We investigated independent single and double knockout cell lines as well as wild-type clones that had gone through the CRISPR/Cas9 procedure to avoid clonal bias for subsequent experiments. Similar to Lin37 knockouts, Rb single knockout cells proliferated normally. In contrast, proliferating double-negative cells exhibited a significant decrease of cells in $G_1$ accompanied by an increase of S and $G_2$/M populations (*Figure 9B*), suggesting a reduced length of the $G_1$ phase. Using both serum deprivation and density growth inhibition, we found that cells depleted of either Lin37 or Rb arrested in $G_0$. In contrast, Lin37/Rb double knockout (DKO) cells displayed a highly significant decrease of $G_0$/$G_1$ cells and an increase of S and $G_2$/M phase cells indicative for the loss of the restriction point (*Figure 9B*). Furthermore, we measured expression of $G_1$/S and $G_2$/M cell cycle genes in these cell lines (*Figure 9C*). Loss of Lin37 led to a significant up-regulation of gene expression in density-arrested and serum-starved cells. A less pronounced de-repression was detected in Rb-negative cells. Cell cycle gene expression in $Lin37^{-/-}/Rb^{-/-}$ cell clones, however, was most significantly up-regulated and generally unchanged between proliferating and density-arrested or serum-starved cells. Additionally, we performed time course experiments with wild-type, single, and double knockout cell lines after serum starvation and re-stimulation (*Figure 9D*). In all mutant cell lines, expression of the $G_2$/M cell cycle gene *Bub1* and the $G_1$/S cell cycle gene *Mcm5* was elevated during $G_0$ and $G_1$. Expression was almost at its maximum in DKO cells immediately after the serum deprivation period, and the $Lin37^{-/-}$ and $Rb^{-/-}$ single knockout lines exhibited an intermediate phenotype with a more pronounced de-repression in the $Lin37^{-/-}$ cells. These observations are consistent with the DNA content of the cells at the given time points, showing that $Lin37^{-/-}/Rb^{-/-}$ cells largely lose their potential to respond to serum starvation and re-stimulation (*Figure 9—source data 1*). In contrast, wild-type and single knockout cells displayed a clear enrichment of 2 n cells after the serum starvation period followed by accumulation of S phase and $G_2$/M cells upon re-stimulation. Thus, combined ablation of both Lin37 and Rb impaires cell cycle arrest following growth-inhibiting conditions.

To test whether the observed de-regulation of cell cycle genes was specific to the loss of Rb or Rb/Lin37, we performed rescue experiments with the same cell lines used for the time course experiments shown in *Figure 9D*. In serum-starved $Rb^{-/-}$ cells, *Cdc45* (maximum expression in $G_1$/S) and *Ttk* (maximum expression in $G_2$/M) wild-type promoters were de-repressed, even if the activity of the *Ttk* promoter was still 10-fold lower than the corresponding CHR promoter mutant (*Figure 10A*). This result is consistent with the observations that cell cycle-dependent expression of *Ttk* is regulated through a CHR site that binds DREAM but not Rb (*Müller et al., 2016*) and that loss of Rb leads to an activation of $G_2$/M cell cycle genes even if it does not bind to their promoters (*Figure 9*). Repression of both promoters could be rescued by expression of wild-type Rb, but not by a mutant that is unable to bind E2f proteins (Rb del22). In Lin37/Rb double knockout cells, both promoters were completely de-repressed in quiescence (*Figure 10B*). Expression of Lin37 or Rb could only partially rescue repression with a more pronounced effect of Rb on the *Cdc45* promoter that binds DREAM and Rb and of Lin37 on the *Ttk* promoter that binds only DREAM. Combined expression of both proteins completely rescued promoter repression after serum deprivation.

Taken together, our data suggest that both Lin37 and Rb are necessary for proper cell cycle gene regulation. Their combined loss leads to de-regulation of cell cycle genes and an inability of cells to respond with cell cycle arrest to growth-inhibiting conditions.

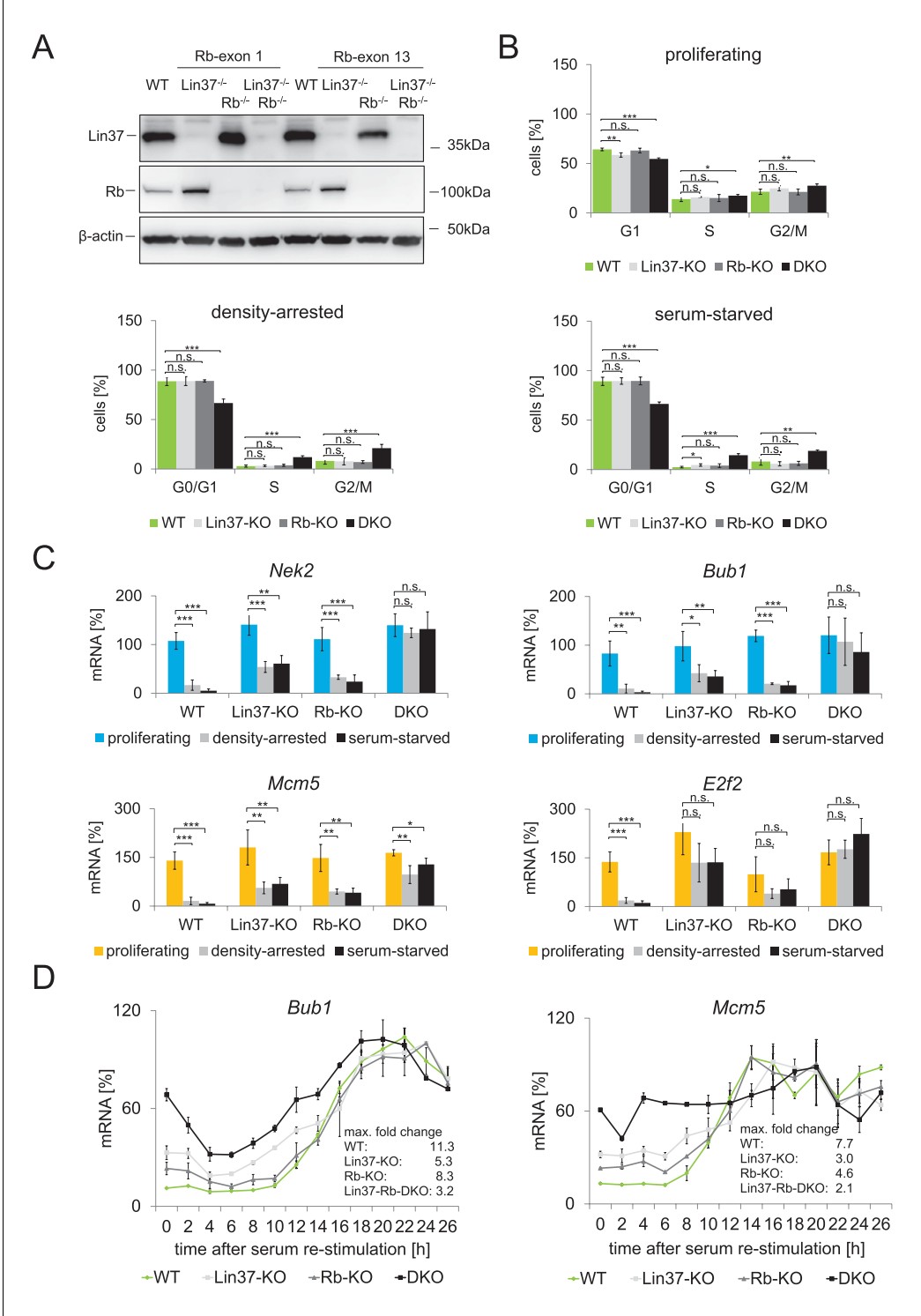

**Figure 9.** Combined ablation of Lin37 and Rb leads to loss of cell cycle control. (**A**) Mutations in the *Rb1* gene of NIH3T3 wild-type and *Lin37*[−/−] cells were introduced by CRISPR/Cas9 nickase by targeting exon 1 or exon 13. Loss of Rb and Lin37 protein expression was confirmed by Western blot. One representative clone for each targeting approach is shown. (**B**) Cell cycle distribution of proliferating, density-arrested and serum-starved wild-type (WT), *Lin37*[−/−] (Lin37-KO), Rb[−/−] (Rb-KO), and *Lin37*[−/−]/*Rb*[−/−] (Lin37-Rb-DKO) cell lines was analyzed by DNA-PI staining. Percentage of cells in $G_0/G_1$, S, and $G_2/M$ cell cycle phases and mean values ± SD of 4 independent cell lines each are given. Significances were calculated by the Students T-Test. (*$p \leq 0.05$, **$p \leq 0.01$, ***$p \leq 0.001$). (**C**) Expression of $G_2/M$ (*Nek2*, *Bub1*) and $G_1/S$ (*Mcm5*, *E2f2*) genes in the same cell lines was

*Figure 9 continued on next page*

*Figure 9 continued*

analyzed by qPCR. For each gene, mRNA levels were normalized to the expression of proliferating wild-type cells which were set to 100% and mean values ± SD are shown. (D) Wild-type (WT), *Lin37*[−/−] (Lin37-KO), *Rb*[−/−] (Rb-KO), and *Lin37*[−/−]/*Rb*[−/−] (DKO) cell lines were arrest by serum starvation in $G_0$ and stimulated to re-enter the cell cycle. mRNA expression of late (*Bub1*) and early (*Mcm5*) cell cycle genes was measured by qPCR at given time points after serum re-stimulation. Mean values ± SD of one representative experiment with two technical replicates for each time point are shown. All data points for each gene were normalized to the maximum mRNA expression of the Lin37 wild-type cell line which was set to 100%. See *Figure 9—source data 1* for cell cycle analysis.
DOI: https://doi.org/10.7554/eLife.26876.013

The following source data is available for figure 9:

**Source data 1.** Cell cycle analysis of synchronized wild-type (WT), *Lin37*[−/−], *Rb*[−/−], and *Lin37*[−/−]/*Rb*[−/−] NIH3T3 cells.
DOI: https://doi.org/10.7554/eLife.26876.014

## Discussion

In our study, we analyze the function of Lin37 and show that it is an essential component of the DREAM repressor complex, but not of the transcriptional activators B-Myb-MuvB and Foxm1-MuvB. Knockout of Lin37 leads to a complete loss of DREAM function. However, the mechanism beyond the key function of Lin37 in transcriptional repression by DREAM is still unclear. The pocket proteins p130/p107 can recruit chromatin-modifying enzymes (*Classon and Dyson, 2001*) which may support a compact nucleosome structure leading to gene repression. Combined knockdown of both proteins leads to a loss of DREAM function (*Litovchick et al., 2007*). Thus, it was believed that p130 and p107 are the central DREAM-specific components mediating gene repression. However, our results provide evidence that p130/p107 cannot repress cell cycle genes when Lin37 is absent or unable to bind to DREAM, although they are still recruited to target gene promoters (*Figure 7*). Given that DREAM can assemble and bind to CHR or E2F sites in the absence of Lin37, it remains to be shown

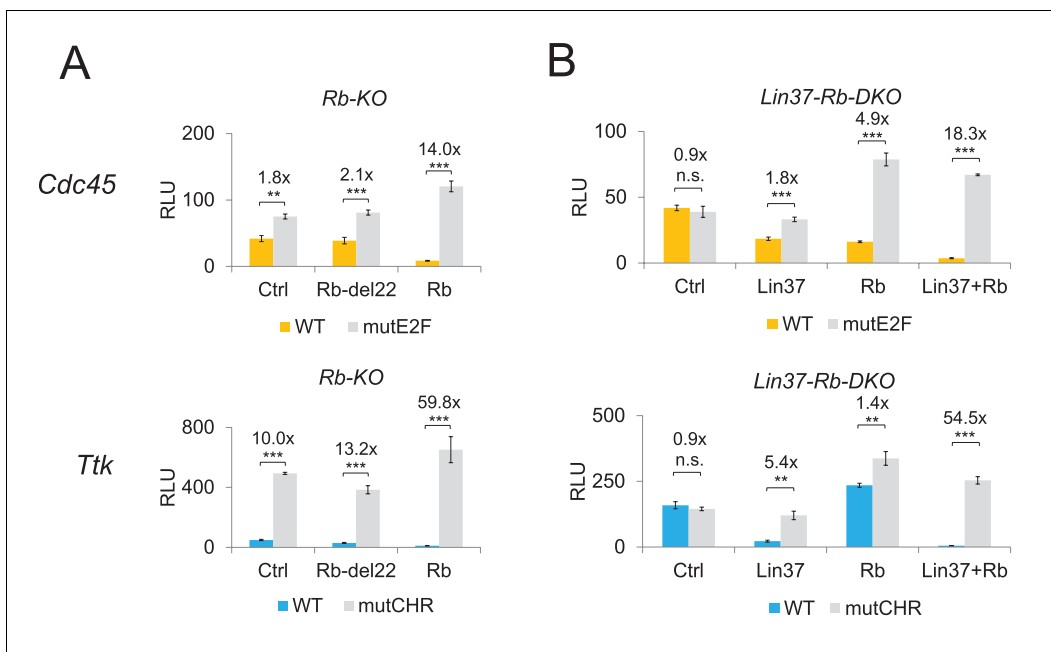

**Figure 10.** Expression of Lin37 and Rb rescues the phenotype of *Rb*[−/−] and *Lin37*[−/−]/*Rb*[−/−] cells. Activities of wild-type (WT) and mutant (mutE2F, mutCHR) $G_1$/S (*Cdc45*) and $G_2$/M (*Ttk*) cell cycle gene promoters in serum-starved (A) Rb[−/−] (Rb-KO) and (B) *Lin37*[−/−]/*Rb*[−/−] (Lin37-Rb-DKO) cells were analyzed by luciferase reporter assays upon co-transfection of plasmids expressing Rb, a Rb mutant with non-functional pocket domain (Rb-del22), Lin37, or of an empty vector (Ctrl). Mean values of relative luciferase light units (RLU) ± SD of three biological replicates are given. Significances were calculated by the Students T-Test (*$p \leq 0.05$, **$p \leq 0.01$, ***$p \leq 0.001$).
DOI: https://doi.org/10.7554/eLife.26876.015

why Lin37 is essential for DREAM-mediated repression. Several models could provide possible explanations: First, Lin37 may recruit yet unknown proteins that are essential for repression by DREAM – most likely chromatin modifiers. Second, loss of Lin37 binding may render DREAM inactive because of conformational changes, which may prevent proper positioning of the complex or inhibit binding of proteins to DREAM without directly contacting Lin37. Third, loss of Lin37 could change post-translational modifications at other DREAM components, possibly p130/p107, that are necessary for function of the complex.

Given that knockdown of DREAM components led to only a moderate (*Litovchick et al., 2007*) or no (*Schmit et al., 2007*) de-repression of cell cycle gene expression in serum-starved cells, it was suggested that DREAM may act redundantly with other repressor complexes like NuRD to repress genes in quiescence. Furthermore, many experiments have shown that cell cycle gene repression and binding of DREAM coincide (*Litovchick et al., 2007*; *Müller et al., 2014*; *Müller et al., 2016*; *Müller et al., 2012*; *Fischer et al., 2013*; *Fischer et al., 2014*; *Fischer et al., 2015*; *Quaas et al., 2012*), but these experiments did not directly address the role of DREAM as a repressor. In contrast, the results presented here provide strong evidence that DREAM is an essential transcriptional repressor to induce quiescence. In serum-starved and density-arrested $Lin37^{-/-}$ cells, expression of cell cycle genes is highly up-regulated. This de-repression is not limited to $G_2/M$ cell cycle genes that are bound by DREAM through CHR promoter elements, but includes also $G_1/S$ cell cycle genes containing E2F binding sites, which also recruit repressing E2f/Rb complexes. Hence, in quiescent cells, loss of DREAM function cannot be compensated by E2f/Rb or by chromatin-modifying complexes. Importantly, promoters become activated in Lin37 knockout cells to a degree that is comparable with mutated promoters containing non-functional CHR sites (*Figures 4* and *10*). Thus, it is unlikely that other proteins binding to CHR sites independent of DREAM contribute to repression. We conclude that DREAM is an essential repressor complex that shuts down expression of both $G_1/S$ and $G_2/M$ cell cycle genes during quiescence.

We observed an up-regulation of more than 300 cell cycle genes in $Lin37^{-/-}$ cells under serum starvation conditions (*Figure 6*), but these cells largely kept their potential to arrest in response to growth inhibiting factors (*Figures 3* and *9*, *Figure 9—source data 1*). This phenotype corresponds to that observed in $p107^{-/-}/p130^{-/-}$ (*Hurford et al., 1997*; *Dannenberg et al., 2000*; *Classon et al., 2000*) and $E2f4^{-/-}/E2f5^{-/-}$ (*Gaubatz et al., 2000*) cells. Given that the loss of all three pocket proteins resulted in an inability of cells to arrest in response to growth-limiting signals (*Dannenberg et al., 2000*; *Sage et al., 2000*), we asked whether a combined knockout of Lin37 and Rb would mirror this phenotype. Indeed, we found that $Lin37^{-/-}/Rb^{-/-}$ cells exhibit massive defects in cell cycle gene expression and do not arrest in response to serum starvation or contact inhibition (*Figures 9* and *10*, *Figure 9—source data 1*). These results indicate that either DREAM or E2f/Rb is sufficient to induce $G_0/G_1$ arrest, even if many cell cycle genes are de-repressed when only one of both complexes is disturbed. In general, our data in combination with earlier findings suggest that Rb and p130/p107 may regulate gene expression through different mechanisms. While Rb can recruit a large variety of chromatin-modifying enzymes through its LxCxE binding cleft, this domain in p130/p107 is essential for DREAM formation and binds to Lin52 (*Guiley et al., 2015*) making it inaccessible for other proteins. Furthermore, binding of Rb to E2f/Dp complexes is sufficient for repressing target genes, while p130/p107 function depends also on MuvB proteins.

The strong activation of cell cycle gene expression in serum-starved $Lin37^{-/-}$ cells as well as the inability of cells to enter quiescence when both Lin37 and Rb are depleted (*Figure 9*, *Figure 9—source data 1*) indicate that Lin37 may be a tumor suppressor. Furthermore, expression of the transcriptional activators *E2f1*, *E2f2*, *Mybl2/B-myb*, and *Foxm1* is up-regulated in $Lin37^{-/-}$ cells (*Figure 6*). These genes are highly expressed in many cancer types which generally predicts poor patient prognosis (*Gentles et al., 2015*). Moreover, viral proteins like HPV E7 and SV40 large T antigen support transformation of cells and cancer development (*McLaughlin-Drubin and Münger, 2009*) and disrupt DREAM (*Fischer et al., 2017*; *Nor Rashid et al., 2011*; *Pang et al., 2014*) as well as E2f/Rb complexes (*Dyson et al., 1989*). In contrast to Rb, mutations of p130 or p107 are only rarely found in cancer (*Cinti et al., 2000*; *Helin et al., 1997*; *Claudio et al., 2000*). Similar to p130 and p107, to the best of our knowledge, a significant enrichment of tumor-derived mutations inactivating Lin37 function has not been identified, even if several amplification, deletion, and mutation events could be detected in some cancer studies (www.cbioportal.org; http://cancer.sanger.ac.uk). Furthermore, we checked several gene expression databases to investigate whether Lin37 protein expression is

altered in cancer tissues (http://www.cbioportal.org; http://www.proteinatlas.org; http://medical-genome.kribb.re.kr/GENT; https://omictools.com/gene-expression-database-of-normal-and-tumor-tissues-tool). However, we could not deduce a general trend in Lin37 expression from these data. While some cancer tissues show reduced Lin37 levels compared to normal tissue (e.g. testis, ovary), others show no significant alteration (e.g. skin, lung) or even an upregulated Lin37 expression (e.g. breast, pancreas). In order to investigate whether a reduced Lin37 expression contributes to cancer formation in specific tissues, additional studies such as experiments in a Lin37 knockout mouse model have to be conducted. However, p130 and p107 have been shown to exhibit tumor suppressive functions in Rb null mouse models when loss of p130 and p107 can support tumor development (*Wirt and Sage, 2010*). Considering that p130/p107 and Lin37 are connected through their function in transcriptional repression, Lin37 may be inactivated only in specific types of cancer or be alternatively mutated to p130 or p107.

Taken together, our data reveal that Lin37 is essential for cell cycle gene repression in cellular quiescence and that a combined loss of Lin37 and Rb impairs cell cycle arrest in response to growth-limiting signals. This indicates a role of Lin37 in cell transformation and cancer development.

## Materials and methods

### Cell culture and drug treatment, MTT assay

NIH3T3 (RRID:CVCL_0594) and C2C12 (RRID:CVCL_0188) cells from DSMZ (Braunschweig, Germany), as well as HCT116 (RRID:CVCL_0291) cells (*Bunz et al., 1998*) were cultivated in DMEM supplemented with 10% FCS. Penicillin/Streptomycin (Sigma-Aldrich) was added to the growth media of NIH3T3 and HCT116 cells. Cells were tested negative for mycoplasma contamination by PCR with a mixture of primers that have been described earlier (*Uphoff and Drexler, 2002*), and identity of the parental cell lines was confirmed by STR profiling (Creative Bioarray). Cells containing pRTS episomal plasmids were selected with Hygromycin B (Sigma-Aldrich) at a concentration of 200 µg/ml. Expression of wild-type Lin37, the CD1+2 MuvB binding mutant, or luciferase together with GFP was induced by adding doxycycline (Sigma-Aldrich) at a concentration of 500 ng/ml. Induced cells expressing GFP were isolated by flow cytometry sorting. Cells were synchronized in $G_0$ by serum starvation (0% FCS) for 60–72 hr or by density-arrest for 5 days after reaching confluency. For cell cycle analyses, NIH3T3 cells were stimulated to re-enter the cell cycle with 20% FCS after the serum deprivation phase. For MTT assays, 2000 cells each were plated in 12 well plates. Starting 24 hr after plating (day1), MTT reagent (Sigma-Aldrich) was added at a final concentration of 0.5 µg/ml and cells were incubated for additional 3 hr before media was removed and formazan crystals were dissolved by adding 1 ml of DMSO. Absorbance was measured at 595 nm. Experiments were performed in triplicates every 24 hr for 6 days.

### Plasmids

The coding region of mouse Lin37 (741bp, NCBI Reference Sequence: NM_029377.2) was amplified from genomic cDNA extracted from NIH3T3 cells by standard PCR and was cloned either in pEGFP-N3 (Clontech), pcDNA3.1(+)FlagN, or pRTS-1 (*Bornkamm et al., 2005*). Site directed mutagenesis (NLS, CD1, CD2, CD1+2) was performed following the QuikChange protocol (Stratagene). Luciferase reporter plasmids containing promoters of *Cdc45*, *Orc1*, *Ttk*, and *Ccnb2* (*Müller et al., 2016*; *Müller et al., 2012*) and expression plasmids for Rb and Rb-del22 (*Qin et al., 1992*) were described earlier. For CRISPR/Cas9 mediated mutagenesis, the plasmid pX335-U6-Chimeric_BB-CBh-hSpCas9n (D10A) (*Ran et al., 2013a*; *Ran et al., 2013b*) (Addgene plasmid #42335) was applied. Sequences of all oligonucleotides are provided in *Supplementary file 1*.

### Generation of Lin37$^{-/-}$, Rb$^{-/-}$, and Lin37$^{-/-}$Rb$^{-/-}$ cell lines by CRISPR/Cas9 nickase

To reduce off-target effects, a CRISPR/Cas9 nickase system (*Ran et al., 2013a*; *Ran et al., 2013b*) was used for insertion of InDels at the boundaries of intron3/exon4 and intron5/exon6 of the Lin37 gene as well as exon1/intron1 and exon13/intron13 of the Rb1 gene. Guide RNAs were designed with a CRISPR design tool (http://crispr.mit.edu). Oligonucleotides with gRNA sequences were cloned into pX335-U6-Chimeric_BB-CBh-hSpCas9n(D10A) following a protocol published earlier

(*Ran et al., 2013b*). Sequences of the oligonucleotides are provided in *Supplementary file 1*. For all CRISPR-Cas9 mediated genome-editing experiments, cells were transfected with 20 µl GeneJuice (EMD Millipore) in 10 cm plates using 4 µg of each px336-sgRNA-plasmid and 1 µg of pMACSk.kII (Miltenyi Biotec). Transfected cell were isolated 48 hr later with the MACSelect Kk – Transfected Cell Selection Kit (Miltenyi Biotec), diluted to 2 cells / 100 µl and plated into 96-well plates at 2 cells per well in conditioned media. Single cell clones were tested by Western blot for expression of Lin37 or Rb. Lin37 knockout NIH3T3 lines used for rescue experiments (Lin37-exon 6: clone 632–2, and Lin37-exon 4: 411–27) were genotyped using PCR, pGEM-T cloning and Sanger sequencing. *Lin37$^{-/-}$Rb$^{-/-}$* were created by inserting additional InDels in the *Lin37$^{-/-}$* clone 632–2. To avoid clonal bias, several knockout cell lines each were analyzed and compared to wild-type clones treated with identical techniques.

## Co-Immunoprecipitations

Lin37 variants (wild-type, NLS, CD1, CD2, CD1+2) with C-terminal EGFP tags or N-terminal FLAG tags were purified from whole cell extracts of transfected HCT116 cells (EGFP constructs) or NIH3T3 cells (FLAG constructs) prepared with IP lysis buffer (*Matsushime et al., 1994*) using the µMACS GFP or DYKDDDDK Isolation Kits (Miltenyi Biotec) following manufacturer's instructions.

## DNA affinity purifications

DREAM components were purified from nuclear extracts of density arrested NIH3T3 *Lin37$^{+/+}$* or *Lin37$^{-/-}$* cells with DNA probes as described earlier and detected by Western blot (*Müller et al., 2012*).

## SDS-PAGE and western blot

SDS-PAGE and western blot were performed following standard protocols as described earlier (*Kirschner et al., 2008*). The following antibodies were applied for protein detection: E2F4 (C-20) (RRID:AB_2097106), p130 (C-20) (RRID:AB_632093), RB (C-2) (RRID:AB_2177334) (Santa Cruz Biotechnology), LIN54 A303-799A (RRID:AB_11218173, Bethyl Laboratories), LIN9 ab62329 (RRID:AB_1269309, Abcam), DP1 Ab-6 (RRID:AB_10985369, Thermo Scientific), GFP-HRP (RRID:AB_247003, Miltenyi Biotec) and FLAG-HRP (RRID:AB_2687602, Miltenyi Biotec), β-actin A5441 (RRID:AB_476744, Sigma-Aldrich), RBBP4 A301-206A (RRID:AB_890631, Bethyl Laboratories), LIN37-T3 (custom-made at Pineda Antikörper-Service, Berlin, Germany) (*Müller et al., 2016*). The monoclonal B-Myb LX015.1 antibody (hybridoma media 1:5) was a kind gift from Roger Watson (*Tavner et al., 2007*).

## Chromatin immunoprecipitation (ChIP)

ChIPs and quantification of promoter fragments by semi-quantitative RT-PCR were performed as described (*Müller et al., 2014*; *Müller et al., 2012*). Sequences of primers are provided in *Supplementary file 1*. The following antibodies were applied: E2F4 (C-20) (RRID:AB_2097106), p130 (C-20) (RRID:AB_632093), LIN54 A303-799A (RRID:AB_11218173, Bethyl Laboratories), LIN37-T1 (custom-made at Pineda Antikörper-Service, Berlin, Germany).

## Semi-quantitative real-time PCR

Total RNA was isolated with TRIzol Reagent (Thermo Fisher Scientific) following the manufacturer's protocol. One-step reverse transcription and quantitative real-time PCR were performed with an ABI 7300 Real-Time PCR System (Applied Biosystems) using the QuantiTect SYBRGreen PCR Kit (Qiagen). For analysis of promoter fragments isolated by ChIP, the reverse transcription step was omitted. See *Supplementary file 1* for primer sequences.

## Next generation sequencing and transcriptome analysis

RNA from serum-starved NIH3T3 cells was extracted with the RNeasy kit (Qiagen). 100 ng of total RNA were fragmented by adding fragmentation buffer (200 mM Tris acetate, pH 8.2, 500 mM potassium acetate and 150 mM magnesium acetate) and heating at 94°C for 3 min followed by ethanol precipitation with ammonium acetate and GlycoBlue (Life Technologies) as carrier. Fragmented RNA was further processed using the Ovation Human FFPE RNA-Seq Library Systems (Nugen) according

to the instructions of the manufacturer. The barcoded libraries were purified and quantified using the Library Quantification Kit - Illumina/Universal (KAPA Biosystems). A pool of up to 10 libraries was used for cluster generation at a concentration of 10 nM using an Illumina cBot. Sequencing of 2 × 100 bp was performed with an Illumina HiScanSQ sequencer at the sequencing core facility of the Faculty of Medicine (University Leipzig) using version three chemistry and flowcell according to the instructions of the manufacturer. RNA-seq data were deposited at the Gene Expression Omnibus (GEO) with the accession number GSE97716.

For all experiments, adapters were clipped using cutadapt 1.8.3 (RRID:SCR_011841) and mapping was performed against mm10 genome applying segemehl version 0-2-0 (RRID:SCR_005494) with split read mapping enabled (*Hoffmann et al., 2009*). Resulting bam files were annotated using rna-counter version 1-5-2 versus gencode.vM5 (https://github.com/jdelafon/rnacounter). Differential expression was computed using edgeR (RRID:SCR_012802) (*Robinson et al., 2010*), after having removed entries for GSAT_MM repeats. A 5% FDR cutoff was used to define sets of significantly differentially expressed genes. For a comparison with DREAM binding data and cell cycle data, mouse genes were mapped to its human homologues using biomart (RRID:SCR_002987) (*Durinck et al., 2009*). Generation of DREAM binding data and cell cycle data sets has been described before (*Müller et al., 2014*). Gene Ontology Analyses were performed with the Gene Ontology online tool (RRID:SCR_002811) (*Gene Ontology Consortium, 2015*; *Ashburner et al., 2000*).

## Luciferase assays

Promoter activities were analyzed by luciferase reporter assays with extracts of transfected serum-starved NIH3T3 or C2C12 cells as described before (*Müller et al., 2012*). For rescue experiments, $Lin37^{-/-}$, $Rb^{-/-}$, and $Lin37^{-/-}/Rb^{-/-}$ NIH3T3 cells were plated in 12-well plates (25,000 cells per well) and transfected by GeneJuice (EMD Millipore) with 150 ng of promoter reporter plasmids along with 200 ng of constructs expressing wild-type or mutant (NSL, CD1, CD2, CD1+2) Lin37 and wild-type or mutant (Rb-del22) Rb, respectively.

## Detection of Lin37-EGFP fusion proteins

NIH3T3 cells were cultivated in 12-well plates on cover slides and transfected with 1.0 µl GeneJuice (EMD Millipore) and 300 ng plasmids expressing wild-type or mutant Lin37 fused to EGFP. Nuclei were stained for 10 min with Hoechst33342 and fluorescence was detected with Structured Illumination fluorescence microscopy, implemented by a Zeiss Apotom 2 (Carl Zeiss, Jena), a HXP120 fluorescence source, an Axiocam 506 mono digital camera, and the Zeiss Zen 2 Software.

## Flow cytometry and cell sorting

The DNA content of NIH3T3 cells was analyzed by staining with propidium iodide (PI) followed by flow cytometry as described earlier (*Müller et al., 2012*). Data was analyzed and figures were created with ModFit LT 5.0. Cells expressing GFP from pRTS episomes were isolated with a BD FACSAria II cell sorter.

## Cell volume measurement

Proliferating and confluent Lin37-KO cells (clone 632–2) were treated with doxycycline (500 ng/ml) to express wild-type Lin37, the CD1+2 MuvB binding mutant or luciferase together with GFP from pRTS episomes. Confluence was maintained over 48 hr. GFP-positive cells were isolated with a BD FACSAria II cell sorter prior to cell volume analysis. Cell volumes were assessed by Coulter counter analysis (Beckman-Coulter Z2 Particle Count and Size Analyzer). For each sample, three technical replicates with approximately 30,000 cells were examined.

## Protein alignment and alignment conservation scores

Lin37 protein sequences of nine deuterostomes were aligned with Clustal Omega (RRID:SCR_001591) (*Sievers et al., 2011*) and amino acid conservation scores were calculated using AL2CO (*Pei and Grishin, 2001*).

## Acknowledgements

The authors thank Kurt Engeland for continued support. We also thank Carola Koschke and Andrea Rothe for technical assistance. Andreas Lösche and Kathrin Jäger performed flow cytometry analyses at the IZKF Leipzig Fluorescence Technologies Core Unit. Fluorescent images were taken at the Biol-maging Core Unit (Leipzig, SIKT). We thank Ronald Weiss for support with the Coulter counter analyses. Knut Krohn, Birgit Oelzner, and Kathleen Schön performed Sanger and Next Generation Sequencing at the IZKF Leipzig DNA Technologies Core Unit. We thank the Engeland and Rubin labs as well as Sunna Hauschildt for helpful discussions and in particular Kurt Engeland, Martin Fischer, Seth Rubin, and Keelan Guiley for critically reading the manuscript. We acknowledge support from the German Research Foundation (DFG) and Universität Leipzig within the program of Open Access Publishing.

## Additional information

### Funding

| Funder | Grant reference number | Author |
| --- | --- | --- |
| Deutsche Forschungsge-meinschaft | MU 3798/1-1 | Gerd A Müller |
| Bundesministerium für Bildung und Forschung | ICGC-Data Mining 01KU1505-C | Stephan H Bernhart |

The funders had no role in study design, data collection and interpretation, or the decision to submit the work for publication.

### Author contributions

Christina FS Mages, Data curation, Formal analysis, Validation, Investigation, Methodology, Writing—review and editing; Axel Wintsche, Data curation, Software, Formal analysis, Methodology, Writing—review and editing; Stephan H Bernhart, Data curation, Software, Formal analysis, Validation, Methodology, Writing—review and editing; Gerd A Müller, Conceptualization, Resources, Data curation, Software, Formal analysis, Supervision, Funding acquisition, Validation, Investigation, Visualization, Methodology, Writing—original draft, Project administration, Writing—review and editing

### Author ORCIDs

Christina FS Mages http://orcid.org/0000-0002-2643-7265
Gerd A Müller http://orcid.org/0000-0002-4967-2487

### Decision letter and Author response

Decision letter https://doi.org/10.7554/eLife.26876.022
Author response https://doi.org/10.7554/eLife.26876.023

## Additional files

### Supplementary files

• Supplementary file 1. Sequences of oligonucleotides used for cloning, mutagenesis, ChIP-qPCR, and reverse transcription qPCR.
DOI: https://doi.org/10.7554/eLife.26876.016

• Supplementary file 2. Transcriptome analysis of quiescent $Lin37^{+/+}$ vs. $Lin37^{-/-}$ cells revel differentially expressed genes.
DOI: https://doi.org/10.7554/eLife.26876.017

• Transparent reporting form
DOI: https://doi.org/10.7554/eLife.26876.018

### Major datasets

The following dataset was generated:

| Author(s) | Year | Dataset title | Dataset URL | Database, license, and accessibility information |
|---|---|---|---|---|
| Mages CFS, Wintsche A, Bernhart SH, Müller GA | 2017 | Lin37 is essential for repression of cell-cycle genes in quiescence | https://www.ncbi.nlm.nih.gov/geo/query/acc.cgi?acc=GSE97716 | Publicly available at the NCBI Gene Expression Omnibus (accession no. GSE97716) |

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
