## [Decision Letter]

Thank you for submitting your article "The DREAM complex through its subunit Lin37 cooperates with Rb to initiate quiescence" for consideration by *eLife*. Your article has been reviewed by three peer reviewers, one of whom is a member of our Board of Reviewing Editors, and the evaluation has been overseen by Tony Hunter as the Senior Editor. The reviewers have opted to remain anonymous.

The reviewers have discussed the reviews with one another and the Reviewing Editor has drafted this decision to help you prepare a revised submission.

The manuscript provides important new insights into the functions of the Lin37 protein, a factor previously shown to be associated with MuvB proteins. The key finding is that Lin37 seems to be essential for repression of cell cycle genes under conditions that generally led to senescence or cell cycle exit in NIH 3T3 cells in the wt situation. Thus it is a co-factor of other pocket proteins in the complex: p130/107. Interesting though not a paradigm shift for repressive complexes in general, in Lin37 null mutations other factors for repression maintain association with endogenous locations but are not active in repression. Furthermore consistent with this point nuclear extracts when incubated with DNA probes for either wt or mutant cells still IP together plus or minus Lin37 in extracts. An important phenotype is that with Lin37 and Rb double null cells cell-cycle arrest is dramatically inefficient. The mechanism of repression for the DREAM complex remains a key problem in the field and the reviewers all agreed that this study presents an important finding that was unexpected and while raising new questions will and should lead to a resolution of this issue. There were only a few major points that need to be dealt with in a revision.

1) A concern for publication in *eLife* is that the work is limited to a single established cell line and thus while solid needs to be – in the opinion of one reviewer – generalized to primary cells or other cell types. The work would be very much enhanced with a CRISPR/CAS knock out in the mouse – particularly with the single Lin 37 and the double rb-/rb; lin37-lin37, which might be embryonic lethal. However, while the mouse work is beyond the present scope, some of the phenotypes, not all, need to be demonstrated in primary cells or another model cell line.

2) The nuclear localization domain studies (Figure 1) add little to the study and one wonders if this experiment indicates that the protein when expressed from recombinant vectors enters with the entire complex into the nuclear space or by itself and exchanges? The issue being in the wt situation does this domain in Lin37 help the entire complex transit or is it even required for function, in an undisturbed setting. That is what would happen with the loss of the NLS in the native gene encoding the Lin 37? The present study could be deleted from the paper or somehow be brought into the major findings with some further experimentation.

3) The authors propose that in the absence of Lin37, cells traverse G1 more rapidly based on their analysis of flow cytometric analysis of DNA content. A similar phenotype was previously reported following the over-expression of cyclin D [https://www.ncbi.nlm.nih.gov/pubmed/8339933]. In that case the cells exhibited a "wee" phenotype with reduced mean cell volume. It would therefore be interesting to know if the Lin37-deficient cells exhibited a similar phenotype. Ideally cell volume is assessed by a Coulter counter analysis of an unsynchronized population of cells. However, forward light scatter in flow cytometry is a common surrogate measurement of cell size that is usually collected at the same time as fluorescence data. Although not essential for the authors' conclusions, I think these data on cell size would be of interest either way and should be easy to obtain either from existing raw cytometry data, or from a Coulter counter analysis.

4) Unless we are mistaken, all of the experiments described were performed with murine NIH 3T3 cells, except for the nuclear localization and co-IPs in Figure 1, panels B and C, which used human HCT116 cells. Ideally these experiments should also have been done in NIH3T3 cells. If not, the authors should at least provide some explanation for this switch. This may not affect the results, but HCT116 are epitheial cells that are wild type for P53 [http://www.pnas.org/content/103/4/976.long], whereas 3T3 cells are mesenchymal cells that are generally mutant for either P53 or for ARF in the same pathway [https://www.ncbi.nlm.nih.gov/pubmed/9393858]. Although it is beyond the scope of this paper, it would be interesting to restore the P53 pathway in the Lin37-deficient 3T3 cells and see if perhaps they are prone to P53-mediated apoptosis.

5) Because the authors are interested in fold-changes in transcription, either of endogenous genes or of reporter genes, the data would be better represented using a log-scale. This makes a two-fold increase equivalent to a two-fold decrease. In addition, the standard errors would then be based on geometric rather than arithmetic means and would likely be more similar at different absolute levels of gene expression.

6) The software used to analyze cell cycle based on DNA content measured by flow cytometry is not really adequate for the task [https://www.ncbi.nlm.nih.gov/pmc/articles/PMC4082777/]. I recommend using software that incorporates cell cycle models, rather than just simple (but arbitrary) linear cutoffs. Modfit is a standard software program for this purpose.

---

## [Author Response]

1) A concern for publication in eLife is that the work is limited to a single established cell line and thus while solid needs to be – in the opinion of one reviewer – generalized to primary cells or other cell types. The work would be very much enhanced with a CRISPR/CAS knock out in the mouse – particularly with the single Lin 37 and the double rb-/rb; lin37-lin37, which might be embryonic lethal. However, while the mouse work is beyond the present scope, some of the phenotypes, not all, need to be demonstrated in primary cells or another model cell line.

We are aware that it has been problematic that our results were based only on NIH3T3 cells deficient for Lin37. We were already in the process of creating Lin37-negative C2C12 cells when we received the decision letter. Following the reviewers’ suggestion, we have added a new paragraph to our manuscript and show that loss of Lin37 has similar effects in C2C12 myoblast cells as we have shown for NIH3T3 cells. Based on several independent C2C12 knockout lines, we show that these cells proliferate normally and arrest in G0, but lose DREAM repressor function since cell cycle genes are de-repressed during quiescence upon Lin37 knockout. Re-expression of Lin37 restores DREAM activity in these cells. Thus, we can show that the phenotype of Lin37-deficient cells is not restricted to fibroblast. Moreover, experiments to create Lin37 knockout mice are in preparation.

2) The nuclear localization domain studies (Figure 1) add little to the study and one wonders if this experiment indicates that the protein when expressed from recombinant vectors enters with the entire complex into the nuclear space or by itself and exchanges? The issue being in the wt situation does this domain in Lin37 help the entire complex transit or is it even required for function, in an undisturbed setting. That is what would happen with the loss of the NLS in the native gene encoding the Lin 37? The present study could be deleted from the paper or somehow be brought into the major findings with some further experimentation.

We respectfully disagree with the comment that the localization studies only add little to our study. In general, as this is the first manuscript providing detailed information on Lin37, we believe that describing the functions of the few conserved domains is of importance.

As the expression of the Lin37 NLS mutant in Lin37-negative cells completely rescues DREAM function (Figure 5), we can assume that the NLS is not essential for the whole complex to enter the nucleus. In addition, we have shown by ChIP that several DREAM components still bind to target gene promoters in Lin37-deficient cells (Figure 7), which supports this observation. Furthermore, since the DREAM-binding deficient variant of Lin37 (CD1+2) is still imported in the nucleus, we know that binding of Lin37 to DREAM is not essential for nuclear import. We hope that, even if we did not perform additional experiments, the reviewers agree with us that the data should be shown.

3) The authors propose that in the absence of Lin37, cells traverse G1 more rapidly based on their analysis of flow cytometric analysis of DNA content. A similar phenotype was previously reported following the over-expression of cyclin D [https://www.ncbi.nlm.nih.gov/pubmed/8339933]. In that case the cells exhibited a "wee" phenotype with reduced mean cell volume. It would therefore be interesting to know if the Lin37-deficient cells exhibited a similar phenotype. Ideally cell volume is assessed by a Coulter counter analysis of an unsynchronized population of cells. However, forward light scatter in flow cytometry is a common surrogate measurement of cell size that is usually collected at the same time as fluorescence data. Although not essential for the authors' conclusions, I think these data on cell size would be of interest either way and should be easy to obtain either from existing raw cytometry data, or from a Coulter counter analysis.

We agree that analyzing the volume of Lin37-deficient cells and testing whether loss of the protein leads to a ‘wee’ phenotype as observed upon over-expression of cyclin D would add helpful information to the manuscript. As the data published in Quelle et al. were obtained from proliferating and contact-inhibited NIH3T3 cells, we performed Coulter counter analyses not only in proliferating knockout and rescue (wild-type and CD1+2 mutant), but also in density-arrested cells. Our data show that the volume of the cells does not change significantly in cells that have lost Lin37 function. The new data are included in Figure 5.

4) Unless we are mistaken, all of the experiments described were performed with murine NIH 3T3 cells, except for the nuclear localization and co-IPs in Figure 1, panels B and C, which used human HCT116 cells. Ideally these experiments should also have been done in NIH3T3 cells. If not, the authors should at least provide some explanation for this switch. This may not affect the results, but HCT116 are epitheial cells that are wild type for P53 [http://www.pnas.org/content/103/4/976.long], whereas 3T3 cells are mesenchymal cells that are generally mutant for either P53 or for ARF in the same pathway [https://www.ncbi.nlm.nih.gov/pubmed/9393858]. Although it is beyond the scope of this paper, it would be interesting to restore the P53 pathway in the Lin37-deficient 3T3 cells and see if perhaps they are prone to P53-mediated apoptosis.

Following the reviewers’ suggestions, we have added additional data to Figure 1 showing that binding of Lin37 depends on the CD1 and CD2 domains also in NIH3T3 cells. Furthermore, we have expressed Lin37 variants with an N-terminal FLAG-tag to show that the C-terminal EGFP-tag used for the experiments in HCT116 cells does not influence binding. In addition, we have tested the localization of Lin37 mutants also in NIH3T3 cells and exchanged the images in Figure 1. The results are consistent with the data acquired in HCT116 cells.

5) Because the authors are interested in fold-changes in transcription, either of endogenous genes or of reporter genes, the data would be better represented using a log-scale. This makes a two-fold increase equivalent to a two-fold decrease. In addition, the standard errors would then be based on geometric rather than arithmetic means and would likely be more similar at different absolute levels of gene expression.

As proposed by the reviewers, we produced versions of the figures using a log-scale. However, we think it is helpful to use linear scales as it is easier to appreciate changes between wild-type and mutants cells on the one hand and between early and late cell cycle phases on the other hand. Thus, we prefer not to change the linear scales of the diagrams.

6) The software used to analyze cell cycle based on DNA content measured by flow cytometry is not really adequate for the task [https://www.ncbi.nlm.nih.gov/pmc/articles/PMC4082777/]. I recommend using software that incorporates cell cycle models, rather than just simple (but arbitrary) linear cutoffs. Modfit is a standard software program for this purpose and a 30-day free trial is available online from the vendor.

As requested, we have re-analyzed all flow cytometry data with ModFit. The data appear in Figure 3, Figure 8, Figure 9, Figure 5—figure supplement 1, and Figure 9—figure supplement 1. Compared to the linear cutoff method, the percentages of G_0_/G_1_, S, and G_2_/M populations have changed in a way that the S phase portions generally appear to be larger. However, these changes do not influence the conclusions drawn from the comparison of wild-type and knockout cells.